# Host-Directed Therapies for Tuberculosis

**DOI:** 10.3390/pathogens11111291

**Published:** 2022-11-03

**Authors:** Eui-Kwon Jeong, Hyo-Ji Lee, Yu-Jin Jung

**Affiliations:** 1BIT Medical Convergence Graduate Program, Kangwon National University, Chuncheon 24341, Korea; 2Department of Biological Sciences, Kangwon National University, Chuncheon 24341, Korea; 3Kangwon Radiation Convergence Research Support Center, Kangwon National University, Chuncheon 24341, Korea

**Keywords:** host-directed therapy, tuberculosis, *Mycobacterium tuberculosis*, drug repositioning

## Abstract

Tuberculosis (TB) is one of the leading causes of death worldwide, consistently threatening public health. Conventional tuberculosis treatment requires a long-term treatment regimen and is associated with side effects. The efficacy of antitubercular drugs has decreased with the emergence of drug-resistant TB; therefore, the development of new TB treatment strategies is urgently needed. In this context, we present host-directed therapy (HDT) as an alternative to current tuberculosis therapy. Unlike antitubercular drugs that directly target *Mycobacterium tuberculosis* (Mtb), the causative agent of TB, HDT is an approach for treating TB that appropriately modulates host immune responses. HDT primarily aims to enhance the antimicrobial activity of the host in order to control Mtb infection and attenuate excessive inflammation in order to minimize tissue damage. Recently, research based on the repositioning of drugs for use in HDT has been in progress. Based on the overall immune responses against Mtb infection and the immune-evasion mechanisms of Mtb, this review examines the repositioned drugs available for HDT and their mechanisms of action.

## 1. Introduction

Host-directed therapy (HDT) aims to treat disease by appropriately modulating the host immune system, not by targeting pathogens. Pathogens express virulence factors for their survival that suppress host immune responses, which can contribute to disease development and progression [1]. However, infection with some pathogens becomes a threat to the host through an overactivation of the host immune response. For example, SARS-CoV-2 causes a cytokine storm, i.e., a systemic inflammatory response, which has been reported to increase mortality in COVID-19 patients [2]. Because HDT can enhance or suppress immune responses depending on its purpose, it can be used for infectious diseases, autoimmune and inflammatory diseases, and cancer treatment [3,4,5].

HDT is particularly important in the treatment of infectious diseases because of antimicrobial resistance (AMR). Infectious diseases caused by viruses or bacteria remain a great threat to mankind, and AMR exacerbates this threat. In 2019, the number of AMR-related deaths worldwide was estimated to be approximately 4.95 million [6], and the discovery of new antibiotics that are effective against antibiotic-resistant bacteria is facing challenges in the 21st century [7]. A strategy that was designed to improve the current circumstances is HDT, which is primarily used as a method to reposition approved drugs for noninfectious diseases, such as cancer, metabolic disorders, and cardiovascular disease. Thus, as a therapeutic strategy, HDT has the potential to rapidly respond to the increasing threat of AMR [8].

Tuberculosis (TB) is an infectious disease caused by *Mycobacterium tuberculosis* (Mtb) and is one of the leading causes of death worldwide. Before the COVID-19 pandemic, TB was ranked first in mortality from a single infectious agent, and it still has a higher mortality rate than HIV/AIDS [9]. Although TB can be treated with anti-TB drugs, the standard regimen for drug-susceptible TB requires a minimum of 4 to 6 months of treatment [10]. The financial burden and side effects associated with long-term treatment lead patients to stop taking the drugs, promoting the development of drug-resistant TB [11]. Drug-resistant TB is resistant to isoniazid (INH) and rifampicin (RMP), the most important first-line anti-TB drugs, and is occasionally resistant to fluoroquinolone and second-line anti-TB drugs, which require a long treatment period and have a significantly reduced cure rate [12]. Since previous exposure to anti-TB drugs is consistently considered a strong risk factor for drug-resistant TB [13], novel therapeutic strategies are needed. In this review, we would like to suggest HDT as an alternative.

## 2. Host Immune Responses against Mtb

### 2.1. Innate Immune Responses

When pathogens invade, hosts execute early defense strategies against pathogens through innate immune responses, and macrophages, dendritic cells (DCs), and neutrophils participate in this first-line defense (Figure 1). They are specialized phagocytes, and their engulfment and degradation of pathogens is the key to innate immunity. Phagocytosis not only sequesters and eliminates pathogens but also is involved in the initiation of adaptive immune responses by presenting the antigen after processing the phagocytosed pathogens. A phagosome containing pathogens cannot clear the pathogens by itself; phagosome maturation, which is a series of processes involving dynamic changes in phagosomal membrane proteins and internal components, must occur for fusion with the lysosome (Figure 2A). Therefore, pathogen recognition, engulfment, and phagosome maturation are required for effective phagocytosis [14]. Macrophages, which function as specialized phagocytes in innate immune responses, play an important role in inflammatory responses and phagocytosis. Inflammation is a defense mechanism of the immune system to remove pathogens and repair damaged tissues [15]. Macrophages that reach the infection site recognize pathogens through pattern recognition receptors (PRRs) on their cell surface. Stimulated PRRs activate transcription factors via intracellular signaling, promoting the expression of proinflammatory cytokines and chemokines (Figure 2B). These factors increase vascular permeability and induce the recruitment of other inflammatory cells, enhancing inflammation and facilitating pathogen clearance. Because inflammatory responses must be tightly regulated, anti-inflammatory cytokines that inhibit and terminate inflammation are important. Anti-inflammatory cytokines attenuate reactivity to interferon-γ (IFN-γ) and decrease antimicrobial effects in macrophages. Appropriately controlled cytokines are beneficial for the host, but because excessive proinflammatory cytokines damage host tissues [16,17] and excessive anti-inflammatory cytokines make the host susceptible to infection, the balance of cytokines is crucial [18].

Mtb is an airborne pathogen that is spread via droplets from individuals with active TB and usually infects the respiratory tract, causing pulmonary TB [19]. In the initial defense against Mtb infection, the innate immune response is dominant, and Mtb entering the respiratory tract encounters innate immune cells, such as alveolar macrophages, neutrophils, and DCs, which participate in first-line defense in the lungs. Mtb is recognized by the PRRs of these cells, activating transcription factors such as NF-κB through downstream signaling to promote the expression of cytokines and chemokines [20]. Tumor necrosis factor-α (TNF-α), a representative proinflammatory cytokine produced primarily by macrophages and DCs, is critical for infection control. Research showed that TNF-α-null mice were susceptible to infection, and patients with inflammatory disorders who were treated with anti-TNF agents had a high tendency for reactivation of latent TB [21,22]. The interleukin-1 (IL-1) family members IL-1α and IL-1β, which are also proinflammatory cytokines, are produced by macrophages, DCs, and neutrophils during Mtb infection and are important in the early defense against infection. Mice lacking IL-1α and IL-1β were shown to be more vulnerable to Mtb infection [23], and mice in which IL-1R, a receptor for IL-1α and IL-1β, had been knocked out showed an increased bacterial burden in the lungs compared with control mice during Mtb infection [24].

Innate immune cells have various strategies to control Mtb invasion, which has been well established through related studies. Macrophages, typical specialized phagocytes, devour and digest Mtb. Through this phagocytosis, macrophages can remove Mtb (Figure 2A) or present antigens to Mtb-specific T cells, initiating adaptive immune responses (Figure 2C). An Mtb-containing phagosome undergoes maturation and fuses with a lysosome to eliminate Mtb inside the cell [25]. Moreover, Mtb virulence factors allow the bacteria to escape phagosomes [26], and Mtb exposed to the cytosol can be removed via autophagy (Figure 2D), which is mediated by STING, parkin, or Smurf1 [27,28]. Parkin and Smurf1, an E3 ubiquitin ligase, are involved in the ubiquitination of cytosolic Mtb, and P62 and NDP52, cargo receptors that recognize ubiquitinated cytosolic Mtb, are recruited, triggering xenophagy, a type of selective autophagy [29,30]. In addition, macrophages produce peptides or chemicals to control Mtb. LL-37, a cathelicidin antimicrobial peptide, displays an antibacterial effect against Mtb in vitro, and the treatment of Mtb-infected mice with LL-37 reportedly reduced bacterial burden in the lungs [31]. Reactive oxygen species (ROS) and reactive nitrogen species (RNS) are highly reactive chemicals that contribute to regulating phagocytosed Mtb (Figure 2B) [32,33]. Neutrophils, which are specialized phagocytes, similar to macrophages, are short-lived but highly recruited in infected areas, and they have granules that can kill bacteria or fungi [34]. Of these granules, azurophil granule proteins have Mtb-killing activity and help macrophages to control intracellular Mtb [35]. Furthermore, DCs act as a bridge connecting innate immunity and adaptive immunity, and NK cells secrete perforin, granzyme, and granulysin to kill Mtb-infected cells and produce IFN-γ, activating macrophages [36].

### 2.2. Adaptive Immune Responses

Macrophages and DCs recruited to the lungs during Mtb infection migrate to secondary lymphoid organs such as lymph nodes, presenting the Mtb antigen to CD4^+^ T cells. CD4^+^ T cells specific for the Mtb antigen are activated into effectors, which contribute to adaptive immunity against Mtb [37]. A study of patients infected with HIV, in whom CD4^+^ T cells are significantly reduced, showed that CD4^+^ T cells play a central role in immunity against Mtb infection. During Mtb infection, Th1 and Th17 cells are primary effector CD4^+^ T cells that produce interferon-γ (IFN-γ) and interleukin-17 (IL-17), respectively (Figure 1) [38]. IFN-γ activates macrophages, promoting phagocytosis, phagosomal maturation, and antigen presentation [39,40]. IL-17 is important for the protective effect induced by BCG vaccination and recruits protective cells, such as neutrophils. However, because excessive IL-17 contributes to immunopathology, such as tissue damage, a balance between Th1 and Th17 responses is important for the control of Mtb growth and the restriction of immunopathology [41]. CD8^+^ T cells, so-called cytotoxic T lymphocytes (CTLs), express perforin, granzyme, and granulysin, and can kill Mtb-infected cells by recognizing the Mtb antigen presented by MHC class I of infected cells. Additionally, CD8^+^ T cells produce cytokines such as IFN-γ and TNF-α [42]. The role of humoral immunity in TB remains unclear (Figure 1). However, since neutralizing antibodies play a crucial role in defense against pathogen invasion in mucosa, and TB is also a disease that occurs in respiratory mucosa, research on Mtb-specific neutralizing antibodies is receiving attention [43]. In fact, antibodies have a protective effect on Mtb [44], and Mtb infection in mice with depleted B cells reportedly increases the bacterial burden in the lungs [45].

### 2.3. Cell Death

Mtb-infected macrophages undergo several types of programmed cell death (PCD) (Figure 2G). Unlike accidental cell death (ACD) such as necrosis, PCD is a process of cell death strictly regulated by intracellular signaling proteins. PCD is divided into non-necrotic cell death, including apoptosis and autophagic cell death, and necrotic cell death, including necroptosis, pyroptosis, and ferroptosis [46]. Because apoptosis and autophagic cell death mostly occur with the membrane intact [47], these cell deaths are known as immunologically silent cell deaths [48]. Conversely, necroptosis, pyroptosis, and ferroptosis cause membrane disruption, inducing inflammation in neighboring cells and tissues [49]. In vitro, while apoptotic cell death is generally beneficial to the host by facilitating Mtb regulation, necrotic cell death is known to be the cell death mechanism exploited by Mtb for dissemination [20].

Apoptosis in macrophages is critical to controlling intracellular bacteria, including Mtb, and it activates innate and adaptive immunity [50,51]. TLRs play an important role in inducing apoptosis in macrophages. According to research, when TLR2 recognizes cell wall components of Mtb in M1 macrophages, the production of ROS and NO in the cytosol increases, which activates the downstream JNK/p53 pathway. Elevated p53 induces proapoptotic signaling molecules, such as p53 upregulated modulator of apoptosis (PUMA), NOXA, and Bcl-2 associated X (BAX), leading to apoptosis [52]. Mtb induces the assembly of NLRP3, one of the NOD-like receptor (NLR) family members, or absent in melanoma 2 (AIM2), a cytosolic DNA sensor, among intracellular PRRs, forming inflammasomes. Inflammasome complexes activate caspase-1, which cleaves pro-IL-1β, pro-IL-18, and the pyroptosis effector gasdermin D (GSDMD). Cleaved GSDMD creates a pore in the membrane, causing cell death and the secretion of mature IL-1β and IL-18 [53,54]. TNF-α induced by Mtb infection provokes necroptosis. When TNF-α stimulates TNF receptor 1 (TNFR1), receptor-interacting serine-threonine kinase (RIPK) 1/3 is activated, and necroptosis effector mixed-lineage kinase domain-like protein (MLKL) forms the membrane pores. Mtb infection induces necroptosis by significantly increasing the expression of the pronecroptotic molecules TNFR1 and Z-DNA binding protein 1 (ZBP1) as well as MLKL [55,56]. Tuberculosis necrotizing toxin (TNT) produced by Mtb inhibits apoptosis and promotes necroptosis in macrophages. TNT degrades cytosolic NAD^+^, and RIPK3 recognizes low levels of NAD^+^ or increased degradation of NAD^+^, promoting necroptosis [57]. Ferroptosis results from either the overaccumulation of free iron or excessive lipid peroxidation. In Mtb-infected cells, the expression of glutathione peroxidase 4 (GPX4), an antioxidant, is downregulated, which leads to an increase in reactive free iron or lipid ROS, resulting in plasma membrane rupture [58].

Neutrophils, which are innate immune cells similar to macrophages, also undergo cell death by Mtb infection. The apoptosis of Mtb-infected neutrophils leads to efferocytosis in which the surrounding phagocytes ingest apoptotic cells, further facilitating the control of Mtb infection [59]. In addition, neutrophils have a unique cell death mechanism called NETosis. As neutrophils die, they release chromatin to form neutrophil extracellular traps (NETs). NETs contain bactericidal proteins that exist in the granules or cytoplasm of neutrophils, and capture Mtb, thus synergizing with phagocytes [60,61].

### 2.4. Metabolism

Mtb-infected macrophages undergo metabolic changes mainly with aerobic glycolysis (Figure 2E), which is related to the M1 polarization of macrophages. As the tricarboxylic acid (TCA) cycle is stopped by increased aerobic glycolysis, succinate accumulates, which stabilizes hypoxia-induced factor-1α (HIF-1α) [62]. HIF-1α is important for immunity involving IFN-γ, and it is essential for the expression of inflammatory cytokines, the production of host-protective eicosanoids, and the regulation of intracellular bacterial replication [63].

Host lipid metabolism also affects immune modulation in Mtb infection. First, in eicosanoid metabolism, prostaglandin E2 (PGE2) mainly functions in the activation, maturation, migration, and regulation of the cytokine secretion of innate immune cells, and it influences both inflammatory and anti-inflammatory responses [64]. PGE2 promotes apoptosis in avirulent Mtb infection [65] and protects macrophages through plasma membrane repair in virulent Mtb infection [66]. Lipoxin A4 (LXA4), another eicosanoid, has both proresolving and anti-inflammatory properties, attenuates M1 polarization, and promotes the M2 polarization of macrophages [67]. The deletion of 5-lipoxygenase to inhibit LXA4 production in mice upregulates IL-12 and IFN-γ, induces nitric oxide synthase (iNOS), and diminishes the bacterial burden in the lungs [68]. Second, Mtb-infected macrophages increase lipid droplet formation in the cytosol (Figure 2F). Mtb infection downregulates the expression of enzymes associated with the TCA cycle, resulting in citrate accumulation [69]. Citrate transported to the cytosol through the citrate carrier is converted into acetyl-CoA, mevalonate, and malonyl-CoA, which synthesize eicosanoid, cholesterol, and free fatty acids, respectively. Among them, cholesterol and free fatty acids are stored in intracellular lipid droplets [70]. In another pathway, Mtb-infected macrophages increase the expression of WNT6, and WNT6 upregulates acetyl-CoA carboxylase 2 (ACC2) via signal transduction, inhibiting mitochondrial fatty acid oxidation. Consequently, macrophages become foamy macrophages with excessive lipid droplet accumulation in the cytoplasm, losing control of Mtb infection and instead providing a nutrient-rich environment for Mtb [71,72].

The role of micronutrients, such as vitamins, in host antibacterial immunity has attracted growing attention. During Mtb infection, vitamin D attenuates intracellular Mtb growth in vitro [73]. Specifically, 1,25-dihydroxy vitamin D3, a biologically active form of vitamin D, inhibits both HIV-1 and Mtb replication through an autophagic mechanism in coinfected macrophages [74]. Vitamin A displays antimycobacterial activity by promoting cholesterol reduction and lysosomal acidification in Mtb-infected monocytes [75] and contributes to Mtb clearance by enhancing autophagy in macrophages [76]. Furthermore, vitamin C has been reported to increase ROS production, which can kill Mtb through the Fenton reaction. In the Fenton reaction, ferrous ions reduced from ferric ions in the presence of reductants react with hydroxyl peroxide to form hydroxyl radicals. One of the reductants participating in this reaction is vitamin C [77]. However, ROS generated by excessive Fenton reactions can cause ferroptosis by forming lipid peroxides on the cell membrane, and as mentioned above, glutathione and GPX enzymes are required to prevent ferroptosis. Vitamins B1 and B5 are involved in Mtb infection control by modulating the host’s proinflammatory responses [78,79].

## 3. Potent HDT Drugs for TB

### 3.1. Modulating Innate Immune Responses

Mtb has immune evasion strategies that can inhibit phagosomal maturation through its virulence factors. For example, protein kinase G (PknG) downregulates host protein kinase C-α (PKC-α), interfering with phagosome–lysosome fusion [80], and phosphatase A (PtpA) inhibits the recruitment of vacuolar H^+^-ATPase (v-ATPase), blocking phagosome acidification [81]. Therefore, enhancing phagosome maturation can be a strategy for TB treatment. Host Abelson tyrosine kinase (Abl) is known to affect lysosomal trafficking [82]. Imatinib, a chemical inhibitor of Abl, which is approved for the treatment of chronic myeloid leukemia (CML), induces the expression and recruitment of the v-ATPase pump subunit, promoting phagosomal acidification and enhancing Mtb killing in human macrophages (Table 1) [83].

Mtb can penetrate the phagosomal membrane via its ESAT-6 secretion system-1 (ESX-1), and autophagy is activated to remove Mtb that has escaped into the cytosol [103]. Thus, autophagy can be a target of HDT for TB. Recent studies have reported that some PRRs activate autophagy, improving host defense against pathogens [104]. Imiquimod, a TLR7 agonist that is a treatment for superficial basal cell carcinoma [84], can regulate intracellular Mtb by increasing autophagy activation and NO production in mouse macrophages (Table 1) [85]. Studies have indicated that isoniazid and pyrazinamide, conventional anti-TB drugs, can also activate autophagy in both human and mouse macrophages. They increase ROS production through NADPH oxidase 2 (NOX2), and activate AMPK via intracellular Ca^2+^ influx, promoting Ca^2+^-dependent autophagy (Table 1) [86]. Likewise, carbamazepine, an anticonvulsant, activates AMPK, inducing mammalian target of rapamycin (mTOR)-independent autophagy during Mtb infection in human macrophages and zebrafish models (Table 1) [87]. Although not approved, several plant-derived natural compounds can also control Mtb infection by activating autophagy. For instance, pasakbumin A, which is a natural compound derived from *Eurycoma longifolia*, triggers autophagy through ERK1/2-mediated signaling, and induces phagosome maturation, contributing to Mtb infection control in mouse macrophages (Table 1) [88]. In addition, resveratrol, a polyphenol commonly found in red wine, is known to extend the lifespan of cells by stimulating the sirtuin 1 (SIRT1)-dependent deacetylation of p53 [105], and recent studies have shown that resveratrol is effective in controlling Mtb infection. During Mtb infection, resveratrol restricts intracellular Mtb growth and induces autophagy and phagosome–lysosome fusion by enhancing the activity of SIRT1 in THP-1 and hMDM cells (Table 1) [89]. Another natural compound extracted from the bark of the *Magnolia* genus, honokiol [106], reinforces autophagy and antimicrobial responses and reduces mitochondrial damage and oxidative stress by activating SIRT3 in Mtb-infected murine macrophages (Table 1) [90].

The inflammatory response is crucial for controlling pathogens, but excessive inflammation damages tissues and is instead detrimental to the host [107]. Therefore, modulating the balance of inflammatory responses is critical for TB treatment. The phosphodiesterase-4 (PDE-4) inhibitor CC-11050, a thalidomide derivative approved for the treatment of leprosy and cancer, was found to downregulate TNF-α, alleviate inflammation, and boost the TB treatment effect of isoniazid in mouse and rabbit models [91,92]. Prednisone and dexamethasone, which are glucocorticoid anti-inflammatory agents, were also reported to reduce the inflammation and ameliorate the immunopathology caused by Mtb via their broad anti-inflammatory actions in human TB patients (Table 1) [93].

### 3.2. Modulating Adaptive Immune Responses

Antigen-presenting cells (APCs) phagocytose antigens and load them onto MHC class I or II molecules, presenting antigens to T cells to initiate adaptive T cell responses. Mtb interferes with this step by preferentially infecting macrophages rather than DCs, which have a more effective and stronger antigen-presenting ability, and this is one of the immune evasive strategies of Mtb [108]. Furthermore, Mtb downregulates the expression of CIITA, a major positive regulator of MHC class II molecules, to prevent the production of MHC class II in macrophages [109]. Therefore, properly activating APCs is critical for adaptive immunity against Mtb infection. G1-A4, a polysaccharide derived from *Tonispora cordifolia*, is a TLR4 agonist that upregulates MHC class II proteins and CD86 and increases the secretion of NO and proinflammatory cytokines, such as TNF-α, IL-6, IL-12, and IFN-γ, in a mouse model (Table 1) [94].

TB is a chronic infectious disease, and when immune stimulation persists for a long time, T cells and APCs upregulate inhibitory receptors, such as programmed death-1/programmed death-ligand 1 (PD-1/PD-L1) [110]. Signaling with PD-1 reduces T cell proliferation and cytokine secretion and induces apoptosis. According to related studies, blocking PD-1 improves the degranulation of CD8^+^ T cells against Mtb, and an increased ratio of IFN-γ-producing lymphocytes was observed in human TB (Table 1) [95].

### 3.3. Targeting Cell Death

Virulent Mtb inhibits the apoptosis of host cells to survive and secure the replication niche in the early phase of infection. After successful intracellular replication, Mtb triggers necrotic cell death to exit the cell and infect new cells [111]. Thus, enhancing apoptosis, which is beneficial for the host, and suppressing necrotic cell death can be a tactic for HDT. TNF-α is essential for controlling Mtb infection; however, excessive TNF-α augments mitochondrial ROS (mtROS) production, and elevated mtROS induces the formation of the mitochondrial permeability transition pore complex (mPTPC), provoking necroptosis through cyclopilin D. In addition, TNF-α activates acid sphingomyelinase, promoting the production of ceramide, a necroptosis inducer. The cyclophilin D inhibitor alisporivir, which is being evaluated for the treatment of hepatitis C in clinical trials, and the acid sphingomyelinase inhibitor desipramine, which is approved for antidepressant use, inhibit the necroptosis caused by ROS through a synergistic effect, relieving tissue damage resulting from excessive TNF-α in a zebra fish model (Table 1) [96]. Ferrostatin-1, a ferroptosis inhibitor, is an antioxidant that scavenges lipid ROS. It has potential for the treatment of Huntington’s disease, periventricular leukomalacia, and renal dysfunction [97]. In Mtb-infected mice, ferrostatin-1 suppresses lipid peroxidation and lowers necrosis and bacterial burden in the lungs (Table 1) [58].

### 3.4. Regulating Metabolism

The accumulation of triacylglycerol and neutral lipids leads to necrosis, facilitating the escape and dissemination of Mtb. Hence, extracellular glucose concentrations can affect the balance between apoptosis and necrosis, and consequently Mtb propagation [70]. According to previous research, Mtb growth increases in diabetes models with hyperglycemia, but macrophages efficiently control Mtb in hypoglycemia models [112]. Metformin is an approved diabetes treatment that reduces excessive blood glucose. In Mtb-infected mice, metformin has been observed to promote phagosome–lysosome fusion by activating AMP-activated protein kinase (AMPK) and obstructing the mitochondrial respiratory chain, and to control BCG by boosting mtROS (Table 1) [98].

Hypercholesterolemia induces spontaneous lipid droplet formation in macrophages, and foamy macrophages that accumulate excess lipid droplets lose their ability to control Mtb [72]. By inhibiting HMG-CoA reductase, statins are used to decrease blood cholesterol levels. In both human and mouse Mtb-infected macrophages, statins lower the cholesterol levels of the phagosomal membrane and promote phagosome maturation and autophagy, affecting Mtb infection control (Table 1) [100].

PGE2 and leukotriene B4 (LTB4), both eicosanoids, are upregulated during Mtb infection [113], and their balance is important to eliminate Mtb without tissue damage. IL-1β signaling stimulates PGE2 production, and PGE2 is necessary to resolve inflammation by LTA4 and LTB4 induced by type I IFN. The PGE2/LTA4 ratio generally decreases in TB [114], and reinforcing PGE2 signaling is consequently beneficial for TB treatment. Zileuton, which is used to treat asthma, inhibits 5-lipoxygenase, blocking leukotriene synthesis. Zileuton reportedly improves the survival rate and decreases bacterial burden and lung damage in Mtb-infected mice (Table 1) [101]. Ibuprofen and acetylsalicylic acid (Aspirin^®^), antipyretic analgesics, are cyclooxygenase (COX) inhibitors that block PGE2 synthesis. Related studies show that these drugs enhance the therapeutic effect of pyrazinamide, a conventional anti-TB drug, in Mtb-infected murine models (Table 1) [102].

## 4. Conclusions

Despite the multifaceted efforts of the World Health Organization (WHO) to end TB, TB remains an infectious disease that threatens humanity. Even though anti-TB drugs are available, the treatment period is too long, and drug treatment is accompanied by side effects. Therefore, curing TB remains a challenge for patients. With the emergence of multidrug-resistant TB (MDR-TB) and extensively drug-resistant TB (XDR-TB), TB is developing into a serious intractable disease. Unlike other pathogens that secrete toxins, Mtb induces strong immune responses in the host, causing host-mediated tissue damage; thus, the immunopathology caused by Mtb is a problem that must be solved. Collectively, HDT has great potential as a therapeutic strategy that can compensate for the defects of current TB therapies, overcome drug resistance, and balance the immune responses of patients. Moreover, because the drugs used for HDT are repositioned drugs that have already been approved and applied in actual clinical practice, this treatment strategy can quickly respond to the rapidly increasing threat of AMR.

Unlike conventional antibiotic therapies, HDT targets the host immune system, enabling personalized therapy according to the patient’s disease condition. In the case of COVID-19, patients who have an excessive inflammatory response are treated with monoclonal antibodies that neutralize inflammatory cytokines such as IL-6 [115], or therapy that removes inflammatory factors in the blood through apheresis [116]. In the case of TB, each patient has a diverse genetic background, and since these individual genetic differences can affect the pathology of TB, endotype-specific HDT based on genetic background can be helpful [117]. For instance, *lta4h*, a gene encoding LTA4 hydrolase, can occur as a single nucleotide polymorphism (SNP) in the promoter region, and the activity of the enzyme varies depending on these genotypes. LTA4 hydrolase catalyzes the production of LTB4, which induces the inflammatory cytokine TNF-α. However, TB patients with the *lta4h* TT genotype secrete excessive TNF due to the overactivation of LTA4 hydrolase, resulting in tissue damage and severe pathology [118]. Metformin, mentioned as a potential HDT drug for TB, promotes phagosome–lysosome fusion via AMPK activation in hMDM, contributing to intracellular Mtb infection control [98]. In a recent study, it was found that during Mtb infection in zebrafish, metformin suppressed the production of TNF-induced mtROS, inhibiting necrosis and contributing to tissue recovery [99,119]. Therefore, metformin may contribute to endotype-specific HDT by alleviating lung pathology while promoting Mtb control in TB patients with the *lta4h* TT genotype.

However, the primary limitation of HDT is that drugs that can be used for the purpose of HDT in clinical practice are currently lacking, even though the safety of these approved drugs has already been demonstrated. One of the reasons for this challenge is that although HDT may be effective in in vitro or in vivo models, the results are different when HDT is applied to actual patients. For example, metformin, which has been reported to be effective in controlling Mtb infection through metabolic regulation, failed to restrict TB in patients with diabetes [120]. In the case of nivolumab and pembrolizumab, which are immune checkpoint inhibitors and can control Mtb by activating CD8^+^ T cells, several reports have indicated that TB becomes more severe or reactivated in cancer patients treated with these drugs [121,122,123,124]. Furthermore, PD-1 blockade increases the susceptibility of the host to Mtb and exacerbates lung pathology because it exaggerates the inflammatory responses through the persistent activation of CD4^+^ T cells [16]. Therefore, HDT might be a revolutionary therapeutic strategy, but since various types of cells in the host react to a single drug, sufficient studies in several animal models must be performed to elucidate their complex interactions, and more case studies have to be conducted before HDT can employed in clinical practice.

## Figures and Tables

**Figure 1 pathogens-11-01291-f001:**
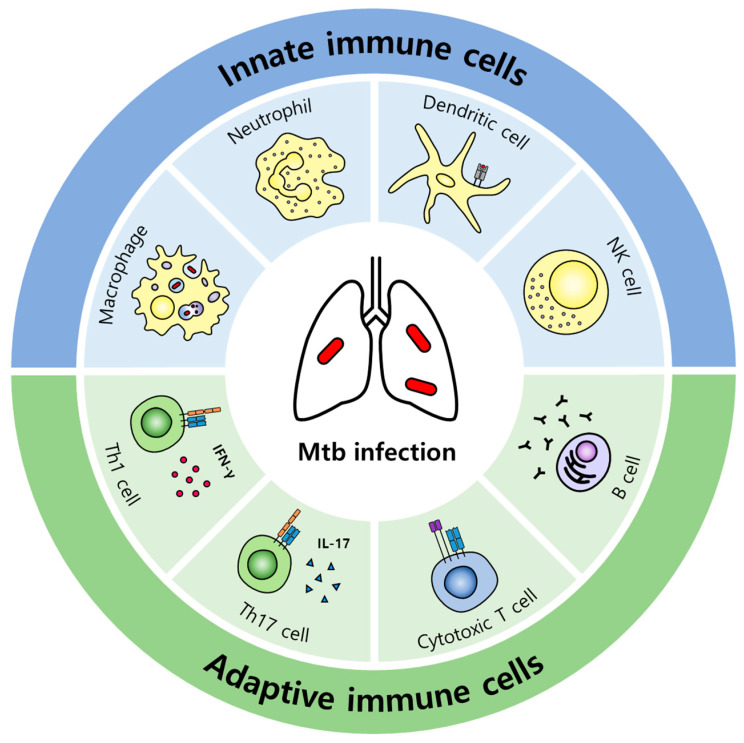
Types of immune cells that contribute to the innate or adaptive immune responses against Mtb infection. Macrophages, neutrophils, and DCs of the myeloid lineage and NK cells of the lymphoid lineage are typically involved in innate immunity against Mtb infection. Specialized phagocytes, macrophages, neutrophils, and DCs can ingest Mtb to eliminate it. When NK cells recognize Mtb-infected cells, they secrete granzyme and perforin inside the granules and induce cell death in infected cells. Because macrophages and DCs also function as antigen-presenting cells, they migrate to the secondary lymphoid organ after phagocytosis and present antigens to CD4^+^ T cells, initiating adaptive immunity. Mtb-specific T cells primarily differentiate into Th1 or Th17 cells that secrete IFN-γ or IL-17, respectively, affecting immune responses against Mtb infection. However, the effect of Th17 cells on immune responses to Mtb remains controversial. Similar to NK cells, Mtb-specific CD8^+^ T cells recognize infected cells and secrete granules, inducing cell death. Humoral immunity associated with B cells and antibodies reportedly contributes to immune responses against Mtb infection, but this process is not yet fully understood.

**Figure 2 pathogens-11-01291-f002:**
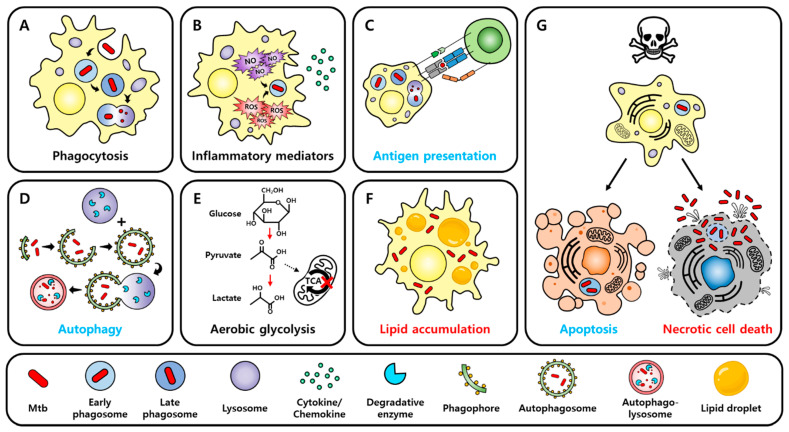
Responses of macrophages to Mtb infection. (**A**) Macrophages ingest Mtb invaded by phagocytosis. Mtb-containing phagosomes become late phagosomes from early phagosomes through phagosome maturation, and they then fuse with lysosomes to eliminate Mtb inside the cell. Although the phagocytosis of macrophages essentially removes pathogens, phagocytosis may be a mechanism that provides refuge to Mtb because Mtb can inhibit phagosome maturation via its virulence factors and can proliferate inside macrophages. (**B**) When macrophages recognize Mtb through PRRs, downstream signaling is triggered and transcription factors are activated, promoting the expression of cytokines and chemokines. Cytokines mediate inflammatory responses, contributing to the control of Mtb infection, and chemokines are responsible for recruiting other immune cells to the infection site. In addition, macrophages produce ROS and RNS, and these reactive chemicals are involved in clearing Mtb inside phagosomes. Inflammation is essential to control pathogen invasion; however, unregulated inflammatory responses due to excessive cytokines cause tissue damage to the host. Thus, the balance of inflammatory responses is crucial in the control of Mtb infection. (**C**) Macrophages, which are antigen-presenting cells, can process phagocytosed Mtb and present its antigen. Macrophages that have ingested Mtb migrate to the secondary lymphoid organs and present antigens to CD4^+^ T cells, triggering adaptive immunity. (**D**) Mtb inside phagosomes can penetrate the phagosome membrane through ESX-1, a virulence factor of Mtb, and be exposed to the cytosol. Autophagy is a mechanism that can eliminate these cytosol-exposed Mtb. Autophagy starts with a phagophore, forms an autophagosome, and then fuses with the lysosome, forming an autophagolysosome. Mtb inside the autophagolysosome is removed by the degradative enzymes of the lysosome. (**E**) Mtb infection promotes the M1 polarization of macrophages. In M1 macrophages, the expression of enzymes associated with the TCA cycle is downregulated, stopping the TCA cycle, and metabolism switches mainly to aerobic glycolysis. Succinate accumulated by the cessation of the TCA cycle stabilizes HIF-1α, and HIF-1α contributes to controlling Mtb infection by promoting the expression of proinflammatory cytokines, among other mechanisms. However, citric acid also accumulates owing to the cessation of the TCA cycle, and excessive citric acid is converted to acetyl-CoA in the cytosol and is involved in the lipid accumulation of macrophages. (**F**) Mtb inhibits the fatty acid oxidation of macrophages and causes an accumulation of lipid droplets in the cytosol. Foamy macrophages with an excessive accumulation of lipid droplets lose their ability to control Mtb, and Mtb can exploit lipid droplets inside foamy macrophages as a nutrient. (**G**) Mtb-infected macrophages undergo various types of cell death. These types of cell death can be divided into non-necrotic cell death (apoptosis) and necrotic cell death. Apoptosis is useful to prevent the propagation of Mtb or to clear Mtb because the membrane is intact during apoptosis. Conversely, necrotic cell death is a membrane-rupturing form of cell death, allowing Mtb to easily spread to neighboring cells. Furthermore, because intracellular materials are leaked during necrotic cell death, inducing inflammatory responses and damage to proximal tissues, necrotic cell death is harmful to the host. Blue text indicates processes that are generally beneficial to the host. Red text indicates mechanisms that are usually exploited by Mtb. Black text indicates mechanisms that generally have two-sided effects.

**Table 1 pathogens-11-01291-t001:** Potential host-directed therapeutics for tuberculosis.

Major Target and Drugs	Mode of Action	Effect	Developmental Stage as HDT for TB	Licensed	Ref.
** *Phagosome maturation and autophagy* **					
Imatinib	induces v-ATPase pump subunit expression and recruitment	promoting phagosomal acidification	preclinical, randomized clinical trial	Leukemia (licensed)	[83]
Imiquimod	stimulates TLR7	activating autophagy and increasing NO production	preclinical	Superficial basal cell carcinoma (licensed)	[84,85]
Isoniazid	activates AMPK through intracellular Ca^2+^ influx, inducing Ca^2+^-dependent autophagy	activating autophagy, increasing NOX2-induced mtROS	preclinical	Anti-TB drug (licensed)	[86]
Pyrazinamide	activates AMPK through intracellular Ca^2+^ influx, inducing Ca^2+^-dependent autophagy	activating autophagy, increasing NOX2-induced mtROS	preclinical	Anti-TB drug (licensed)	[86]
Carbamazepine	induces mTOR-independent autophagy through AMPK activation	activating autophagy	preclinical	Anti-convulsant (licensed)	[87]
Pasakbumin A	ERK1/2-mediated signaling	activating autophagy, inducing phagosome maturation	preclinical	Natural compound (preclinical)	[88]
Resveratrol	activates SIRT1	activating autophagy, inducing phagosome-lysosome fusion	preclinical	Nutritional supplement (licensed)	[89]
Honokiol	activates SIRT3	activating autophagy, enhancing antimicrobial response, reducing mitochondrial damage and oxidative stress	preclinical	Nutritional supplement (licensed)	[90]
** *Inflammation* **					
CC-11050	inhibits PDE-4 and downregulates TNF-α	resolving inflammation, improving therapeutic effect of isoniazid	preclinical	Leprosy, cancer (licensed)	[91,92]
Prednisone, Dexamethasone	Anti-inflammation and immunosuppression	resolving inflammation, attenuating pathology	preclinical	Anti-inflammatory (licensed)	[93]
** *Adaptive immunity* **					
G1-4A	stimulates TLR4	upregulating expression of MHC class II and CD86, increasing secretion of proinflammatory cytokine and NO	preclinical	Natural compound (preclinical)	[94]
Nivolumab, Pembrolizumab	anti-PD-1 mAb	enhancing degranulation of CD8^+^ T cell, increasing ratio of IFN-γ producing lymphocytes	preclinical	Cancer (licensed)	[95]
** *Cell death* **					
Alisporivir	inhibits ROS-induced necroptosis	attenuating tissue damage induced by excessive TNF-α through synergistic effect with Desipramine	preclinical	Hepatitis C (clinical trial)	[96]
Desipramine	inhibits ROS-induced necroptosis	attenuating tissue damage induced by excessive TNF-α through synergistic effect with Alisporivir	preclinical	Antidepressant (licensed)	[96]
Ferrostatin-1	antioxidant that eliminates lipid ROS	inhibiting lipid peroxidation	preclinical	Huntington’s disease, periventricular leukomalacia, kidney dysfunction(preclinical)	[58,97]
** *Metabolism* **					
Metformin	activates AMPK and inhibits mitochondrial respiratory chain	promoting phagosome-lysosome fusion, increasing mtROS	randomized clinical trial	Diabetes (licensed)	[98,99]
Statins	inhibits HMG-CoA reductase	promoting phagosome maturation and autophagy	preclinical	Hypercholesterolemia (licensed)	[100]
Zileuton	inhibits 5-LOX	inhibiting leukotriene synthesis	preclinical	Asthma (licensed)	[101]
Ibuprofen, acetylsalicylic acid	inhibits COX	inhibiting PGE2 synthesis, improving therapeutic effect of pyrazinamide	preclinical	Pain, fever (licensed)	[102]

## Data Availability

Not applicable.

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
