# Peer review of "Host-Directed Therapies for Tuberculosis"

_pathogens, 2022, doi:10.3390/pathogens11111291_

Round 1

Reviewer 1 Report

Abstract: 

Would HDT be an alternative or an adjunct to antibiotics? 

Should the word be repositioned or repurposed? 

Major: 

Line 27-29 is an excellent, succinct summary of the problem: sometimes HDT is in response to anergic immunity and sometimes it needs to be in response to pathologic inflammation. If there are such divergent immune endotypes, how will HDT be guided? Specifically, how will this be implemented in resource constrained environments? The field does not have an answer for this, but it is worth mentioning that COVID has normalized the clinical use of IL6, CRP and therefore it is worth mentioning that this might be needed for guidance of endotype-specific HDT. Also, I did not see a discussion about timing. Probably immunosuppression is beneficial early, while augmenting host immunity is probably better later, after antibiotics have decreased bacillary burden 3-4 logs. 

Line 55: Maybe the authors are right, but I see HDT as an adjuvant to existing antibiotics, not as an alternative. I would be careful with the language here. 

Figure 1: I would remove the thickness of the arrows; as the authors rightly point out, there is not a single immune pathology to TB. Therefore, there is not a single immune cell type that is always beneficial and always detrimental. I think the arrows are misguided and not backed by the clearest evidence. 

Line 92-98: well stated. But please cite the original research and not review articles. Consider Barber JI 2011, Kaufmann & Barber 2021, Nair & Diamond 2018. All of these demonstrate that perturbing immunity to increase pro-inflammatory cytokines results in fatal immune-induced lung pathology

Figure 2: per the red, blue and black color code, can I suggest you add the word “usually?” 

Line 178: macrophages and DCs that first encounter Mtb are tissue resident, not those that are recruited into the lung. 

Section 2.3: It might be worth also mentioning cuproptosis

Line 271: I might consider removing the word “recently.” There are publications related to Vit D and Vit A in TB going back ~100 years. 

Line 278: Could you spend a few sentences linking Vitamin C, ROS, the fenton reaction and glutathione mentioned in the previous paragraph? 

Line 339-344: I would recommend discussing the IFNg induced immune pathology demonstrated by Barber & Sher, 2011 when they infected PD1 KO mice with Mtb. Would this strategy have worked in a model that better recapitulated human disease, by first infecting the mice with Mtb, then decreasing bacillary burden with antibiotics, and then giving an PD1 monoclonal antibody? 

Section 364-371: please discuss the recent 2 studies using metformin and everolimus in TB (Wallis 2022, Lancet Resp and Padmapriydarsini CID 2022. Note that these 2 studies looked at morbidity as the clinical outcome (lung function) rather than just microbiologic cure. Should this be discussed in more depth? In particular, the metformin study demonstrated that a slower time to culture conversion resulted in better radiographic healing. Which is more important? 

In summary, this is a very nice summery of HDT for TB. I’m not sure it adds a lot of new ideas or concepts. My major criticism is that the authors do a nice job of stating that there is not a single endotype leading to immunopathology, but they don’t discuss what this pragmatically means. How do we determine when we need to give prednisone versus when Nivolumab? One inhibits host immunity, while the other dials it up. Are there some TB patients that need prednisone, while others need Nivolumab? Or do all TB patients need prednisone during the first few weeks of antibiotics, and then only a few need Nivolumab during the last few weeks of antibiotics? There are no answers to these questions, but the authors should discuss how the field needs to address these major hurdles.  This is a critical part of the any future HDT for TB 

Author Response

Editorial comment Reviewer 1

 Comments and Suggestions for Authors

Abstract: 

  1. Q) Would HDT be an alternative or an adjunct to antibiotics? 

Authors’ response. In this review, we attempt to describe HDT as an alternative treatment for antibiotic-resistant Mtb infection.

  1. Q) Should the word be repositioned or repurposed? 

 Authors’ response. Drug repositioning and drug repurposing are both used commonly with the same meaning, but to avoid confusion, we have revised the manuscript to consistently use the term as “drug repositioning”. (Page 1: line 40)

Major: 

  1. Q) Line 27-29 is an excellent, succinct summary of the problem: sometimes HDT is in response to anergic immunity and sometimes it needs to be in response to pathologic inflammation. If there are such divergent immune endotypes, how will HDT be guided? Specifically, how will this be implemented in resource constrained environments? The field does not have an answer for this, but it is worth mentioning that COVID has normalized the clinical use of IL6, CRP and therefore it is worth mentioning that this might be needed for guidance of endotype-specific HDT. Also, I did not see a discussion about timing. Probably immunosuppression is beneficial early, while augmenting host immunity is probably better later, after antibiotics have decreased bacillary burden 3-4 logs. 

Authors’ response. This is a legitimate point that the use of IL6 and CRP in COVID-19 patients could be applied to patients with certain endotypes of tuberculosis and is very interesting. According to the reviewer’s suggestion, this information has been added to the discussion of TB endotype of HDT treatment for patients with different endytypes. However, there is insufficient evidence to suggest a practical approach of when HDT could be introduced to patients with different endotypes. Therefore, to suggest or provide guidance on the proper timing of the regulation of host immunity is beyond the scope of this paper, which addresses HDT from a basic immunological point of view. (Page 12: line 518-525, ref. 115-117)

Line 55: Maybe the authors are right, but I see HDT as an adjuvant to existing antibiotics, not as an alternative. I would be careful with the language here. 

Authors’ response. On the basis of the approaches used thus far, we believe that the reviewer's argument could be correct in the context of HDT being applied as an adjuvant to antibiotics. However, in this review, we propose the perspective that HDT can be used as both an alternative to antibiotics and an adjuvant to antibiotics.

Figure 1: I would remove the thickness of the arrows; as the authors rightly point out, there is not a single immune pathology to TB. Therefore, there is not a single immune cell type that is always beneficial and always detrimental. I think the arrows are misguided and not backed by the clearest evidence

Authors’ response. We agree with the reviewer’s point; therefore, Figure 1 has been replaced with a new version without the arrows.

Line 92-98: well stated. But please cite the original research and not review articles. Consider Barber JI 2011, Kaufmann & Barber 2021, Nair & Diamond 2018. All of these demonstrate that perturbing immunity to increase pro-inflammatory cytokines results in fatal immune-induced lung pathology

 Authors’ response. Thank you for the suggestion. We added two references (Ref. 16,17) that an excessive proinflammatory response exacerbates lung pathology. Unfortunately, we were not able to find Kaufmann & Barber 2021.

Figure 2: per the red, blue and black color code, can I suggest you add the word “usually?” 

Authors’ response. The manuscript has been revised according to the reviewer’s suggestion. (Page 5: line 212-213)

Line 178: macrophages and DCs that first encounter Mtb are tissue resident, not those that are recruited into the lung

Authors’ response. We make no claim that tissue resident macrophages and DCs first encounter Mtb in the lung. Therefore, the word “early” has been deleted to avoid confusion. (Page 5: line 216)

Section 2.3: It might be worth also mentioning cuproptosis

 Authors’ response. We thank the reviewer for this comment. We would like to describe cell death in the context of Mtb infection, but it appears that no papers have yet been published on the correlation between cuproptosis and Mtb infection. We are also studying the cell death mechanisms in several infectious models, so we hope to present our cell death results, including those for cuproptosis, as part of another study.

Line 271: I might consider removing the word “recently.” There are publications related to Vit D and Vit A in TB going back ~100 years. 

Authors’ response. The manuscript has been revised accordingly. (Page 7: line 332)

Line 278: Could you spend a few sentences linking Vitamin C, ROS, the fenton reaction and glutathione mentioned in the previous paragraph? 

Authors’ response. The manuscript has been revised accordingly, and a related reference has been added.

(Page 7: line 339-341, Ref. 77)

Line 339-344: I would recommend discussing the IFNg induced immune pathology demonstrated by Barber & Sher, 2011 when they infected PD1 KO mice with Mtb. Would this strategy have worked in a model that better recapitulated human disease, by first infecting the mice with Mtb, then decreasing bacillary burden with antibiotics, and then giving an PD1 monoclonal antibody? 

Authors’ response. In the discussion, the presented article is mentioned while stating the conflicting effect of PD-1 blockage. (Page 12: line 588-590, Ref. 16)

Section 364-371: please discuss the recent 2 studies using metformin and everolimus in TB (Wallis 2022, Lancet Resp and Padmapriydarsini CID 2022. Note that these 2 studies looked at morbidity as the clinical outcome (lung function) rather than just microbiologic cure. Should this be discussed in more depth? In particular, the metformin study demonstrated that a slower time to culture conversion resulted in better radiographic healing. Which is more important? 

Authors’ response. In the conclusion, we presented information on endotype-specific HDT while adding the topic of personalized therapy, and we mentioned metformin as a potential drug for HDT HDT. (Page 12: line 518-536, Ref. 95, 96, 117-119) As the authors are all basic immunologists, it is beyond our knowledge to discuss the importance of balancing clinical symptom relief and microbiological sterilization. Although TB has a challenge in completely eradicating the causative agent, lung damage caused by chronic inflammation is also a serious problem. Even if TB is cured, there is a risk of sequelae due to chronic inflammation, so it is also very important for damaged tissue to recover. In this context, metformin, which promotes intracellular Mtb infection control while simultaneously contributing to tissue repair, is considered a very good example of an HDT drug. Eliminating the disease itself by eradicating the causative agent is the basis of treating infectious diseases, but the goal is also to alleviate the pathology with no sequelae when the patient’s disease is cured.

In summary, this is a very nice summery of HDT for TB. I’m not sure it adds a lot of new ideas or concepts. My major criticism is that the authors do a nice job of stating that there is not a single endotype leading to immunopathology, but they don’t discuss what this pragmatically means. How do we determine when we need to give prednisone versus when Nivolumab? One inhibits host immunity, while the other dials it up. Are there some TB patients that need prednisone, while others need Nivolumab? Or do all TB patients need prednisone during the first few weeks of antibiotics, and then only a few need Nivolumab during the last few weeks of antibiotics? There are no answers to these questions, but the authors should discuss how the field needs to address these major hurdles.  This is a critical part of the any future HDT for TB 

Authors’ response. We thank the reviewer for sharing their opinion on clinical HDT treatment strategies for patients with different endotypes of tuberculosis. We believe that immune enhancement and immunomodulation should be critical for the introduction of HDT in tuberculosis patients. The intensity and duration of immunity to Mtb is important when considering the appro-priate timing of treatment for Mtb as well as the likely genetic background of each patient, but the answer as to the timing of regulating the immune response is unknown, and there is insufficient evidence to suggest a practical approach. However, this review is primarily written from the perspective of addressing the mechanisms of HDT, as evidenced by the results of preclinical studies and aspects of the primary immune response to Mtb infection. Therefore, the sentence has been modified in the last part of the conclusion. (Pages 12: Lines 590-594). It would be appropriate to discuss the future of HDT in another paper after the controversies regarding practical application have been resolved.

Reviewer 2 Report

Review article by Eui-Kwon Jeong et al. discussed about M.tb immune evasion strategies followed  by targeting host immune pathways using repurposed drugs to treat Tuberculosis disease and to overcome antimicrobial resistance in this review article.  

The manuscript was well written; however, this review did not provide the pros and cons towards host directed therapy (HDT). Lacking discussion on future directions of HDT to treat Tb disease. After minor revision this review might have possibility to merit publication in Pathogens Journal. 

Below are suggestions for minor revision of the manuscript:

1.     This review article has given several examples of host immune pathways, which can be targeted for TB therapy, however mouse and human immune cell pathways not distinguished well enough in explaining HDT’s to treat TB disease.

2.     More detailed discussion is needed on immune pathways targeted by HDT drugs.

3.     Pros and cons section is needed on HDT’s.

I have no major concerns with this paper to publish in Pathogens Journal after minor revision.

Author Response

Reviewer2

Comments and Suggestions for Authors

Review article by Eui-Kwon Jeong et al. discussed about M.tb immune evasion strategies followed  by targeting host immune pathways using repurposed drugs to treat Tuberculosis disease and to overcome antimicrobial resistance in this review article.  

The manuscript was well written; however, this review did not provide the pros and cons towards host directed therapy (HDT). Lacking discussion on future directions of HDT to treat Tb disease. After minor revision this review might have possibility to merit publication in Pathogens Journal. 

Below are suggestions for minor revision of the manuscript:

  1. This review article has given several examples of host immune pathways, which can be targeted for TB therapy, however mouse and human immune cell pathways not distinguished well enough in explaining HDT’s to treat TB disease.

 Authors’ response. Following the reviewer’s comments, all experimental models of drugs presented in the main body of the paper have been described in detail in the following text: Page 9-10: line 381, line 383, line 394-395, line 399, line 403, line 419, line 428, line 442, line 448, line 459-460, line 476 etc.

  1. More detailed discussion is needed on immune pathways targeted by HDT drugs.

 Authors’ response. According to the reviewer’s comments, we tried to explain each targeting immune response for each HDT presented in detail as much as possible, and we described a treatment strategy that can be applied differently to each tuberculosis patient in an endotype-specific manner in the Conclusion in this revised manuscript

  1. Pros and cons section is needed on HDT’s.

Authors’ response. The pros and cons of HDT in tuberculosis patients have been discussed in the Conclusion section (Page 11-12: Lines 518 to 594).

Round 2

Reviewer 1 Report

Previous comments addressed; no further comments